# Evolutionarily conserved brainstem architecture enables gravity-guided vertical navigation

**Yunlu Zhu, Hannah Gelnaw, Franziska Auer, Kyla R. Hamling, David E. Ehrlich, David Schoppik** [ORCID] *

Departments of Otolaryngology, Neuroscience & Physiology, and the Neuroscience Institute, New York University Grossman School of Medicine, New York, New York, United States of America

* schoppik@gmail.com

**Data Availability Statement:** All raw data and analysis code can be downloaded at the Open Science Framework DOI: 10.17605/OSF.IO/AER9F.

## Abstract

The sensation of gravity anchors our perception of the environment and is important for navigation. However, the neural circuits that transform gravity into commands for navigation are undefined. We first determined that larval zebrafish (*Danio rerio*) navigate vertically by maintaining a consistent heading across a series of upward climb or downward dive bouts. Gravity-blind mutant fish swim with more variable heading and excessive veering, leading to less effective vertical navigation. After targeted photoablation of ascending vestibular neurons and spinal projecting midbrain neurons, but not vestibulospinal neurons, vertical navigation was impaired. These data define a sensorimotor circuit that uses evolutionarily conserved brainstem architecture to transform gravitational signals into persistent heading for vertical navigation. The work lays a foundation to understand how vestibular inputs allow animals to move effectively through their environment.

## Introduction

Animals adopt navigational strategies tailored to their sensory ecology [1,2]. Perception of the environment, particularly for species that swim or fly [3–6], is anchored by the sense of gravity [7,8]. Vertebrates use otolithic organs in the vestibular system of the inner ear to transduce linear acceleration due to gravity [9]. Vestibular information has long been thought to impact spatial navigation [10], shaping behaviors such as stabilization of vision and posture, perception of self-motion and head direction, motor coordination, and path integration [8,11–13], and can modulate the activity of neurons responsible for navigation, such as head direction cells of the mammalian limbic system [14–16]. However, complete circuits that transform sensed gravity into motor signals for navigation remain undefined.

The larval zebrafish (*Danio rerio*), a small translucent vertebrate, is an ideal model to discover neural substrates for gravity-guided navigation [17–21]. Zebrafish larvae translate in short discrete bouts punctuated by periods of inactivity. Their separable active and passive phases of locomotion facilitate dissection of the neural-derived commands for movement from biomechanical consequences. Zebrafish maintain a dorsal-up orientation relative to

**Funding:** Research was supported by: The National Institute on Deafness and Communication Disorders of the National Institutes of Health under award numbers R01DC017489 (DS), and F31DC019554 (KRH). URL: https://www.nidcd.nih.gov The National Institute of Neurological Disorders and Stroke under award numbers, T32NS086750 (KRH) and R61NS125280 (DS). URL: https://www.ninds.nih.gov The Leon Levy Fellowship in Neuroscience from the Leon Levy Foundation (YZ). URL: https://leonlevy.org The Rainwater Tau Leadership Fellowship from the Rainwater Charitable Foundation (YZ). URL: https://rainwatercharitablefoundation.org The funders had no role in study design, data collection and analysis, decision to publish, or preparation of the manuscript.

**Abbreviations:** dpf, days postfertilization; IQR, interquartile range; MAD, median absolute deviation; NPH, nucleus prepositus hypoglossi; SAMPL, Scalable Apparatus to Measure Posture and Locomotion.

gravity using postural reflexes [22–26] and learned control of movement timing [27]. These vestibular behaviors rely entirely on otolithic organs [28], in particular, the gravity-sensing utricle [29–31]. Anatomically, utricle-recipient vestibular nuclei relay information to the spinal cord directly [32–34] and indirectly [24,25,30,35,36] through a highly conserved midbrain population called the interstitial nucleus of Cajal/nucleus of the medial longitudinal fasciculus (INC/nMLF) [37,38]. Finally, larval zebrafish climb and dive in the water column for a variety of reasons. They surface to inflate their swim bladder [39,40], hunt microorganisms in the upper water column [41], and adjust depth following changes to illumination [42–44]. While zebrafish navigation in the horizontal plane has been explored [45–47], it is unclear if they similarly maintain vertical heading to navigate in depth.

Here, we combined high-throughput behavioral analysis of vertical locomotion and loss-of-function assays to explore neural circuits for gravity-guided navigation. We first established that larvae swim in a series of bouts with consistent heading to navigate in the dark. Stable control of heading allowed larvae to effectively change depth. Gravity sensation can modulate this navigation behavior, as mutant fish without utricular otoliths navigate depth poorly, swimming with more variable heading and excessive veering. Lesions of ascending utricle-recipient neurons in the tangential vestibular nucleus recapitulated this phenotype, while lesions of descending vestibulospinal neurons did not. The INC/nMLF receives ascending inputs; lesions of the INC/nMLF disrupted heading and navigation efficacy. Taken together, our data reveals a conserved hindbrain-midbrain-spinal cord circuit that transformed sensed gravity to commands to maintain heading for effective vertical navigation. More broadly, we reveal ancient architecture that leverages sensed gravity to move effectively through the world.

## Results

### Larval zebrafish navigate depth by maintaining a consistent heading over a series of swim bouts

We first examined whether larval zebrafish maintain a consistent heading as they navigate in depth [48]. To measure behavior, we used a high-throughput real time Scalable Apparatus to Measure Posture and Locomotion (SAMPL) [49]. SAMPL records body position and posture in the pitch axis (nose-up/nose-down) as larval zebrafish swim freely in depth (Fig 1A). We examined freely swimming larvae from 7 to 9 days postfertilization (dpf) in complete darkness. We measured the trajectory of swim directions relative to horizontal and observed both upward and downward swim bouts (Fig 1B and 1C), indicating that larvae climb and dive in the water column. To quantify the spread of individual swim directions, we defined variability as the median absolute deviation (MAD) of swim bout trajectories (Fig 1D).

The depth change resulting from a single swim bout was small (0.34 [0.57] mm, median of absolute depth displacement with interquartile range [IQR]), so we hypothesized that larval zebrafish integrate a series of swim bouts to adjust their depth effectively. We quantified and parameterized the statistics of short series of sequential bouts. Directions of consecutive bouts were highly correlated (Fig 1E), determined by the coefficient of determination of direction (Fig 1F), and highly consistent, defined as the slope of the best-fit line between directions of consecutive bout (Fig 1G and Table 1 for parameter definitions and statistics). As the series continued, bout direction became increasingly less correlated with the first bout (Fig 1F and 1G) reaching chance level around the 10th consecutive bout (S1 Fig). To quantify the amount of direction change during consecutive bouts, we defined veering as the mean of absolute direction differences between adjacent bouts (Fig 1H). Compared to shuffled bouts, fish veered significantly less during observed consecutive bouts (Fig 1I; 5.61 [6.85]° versus 23.82 [16.03]°, observed versus shuffled, median with IQR, $P_{median-test} <$ 1e-16), indicating that larval

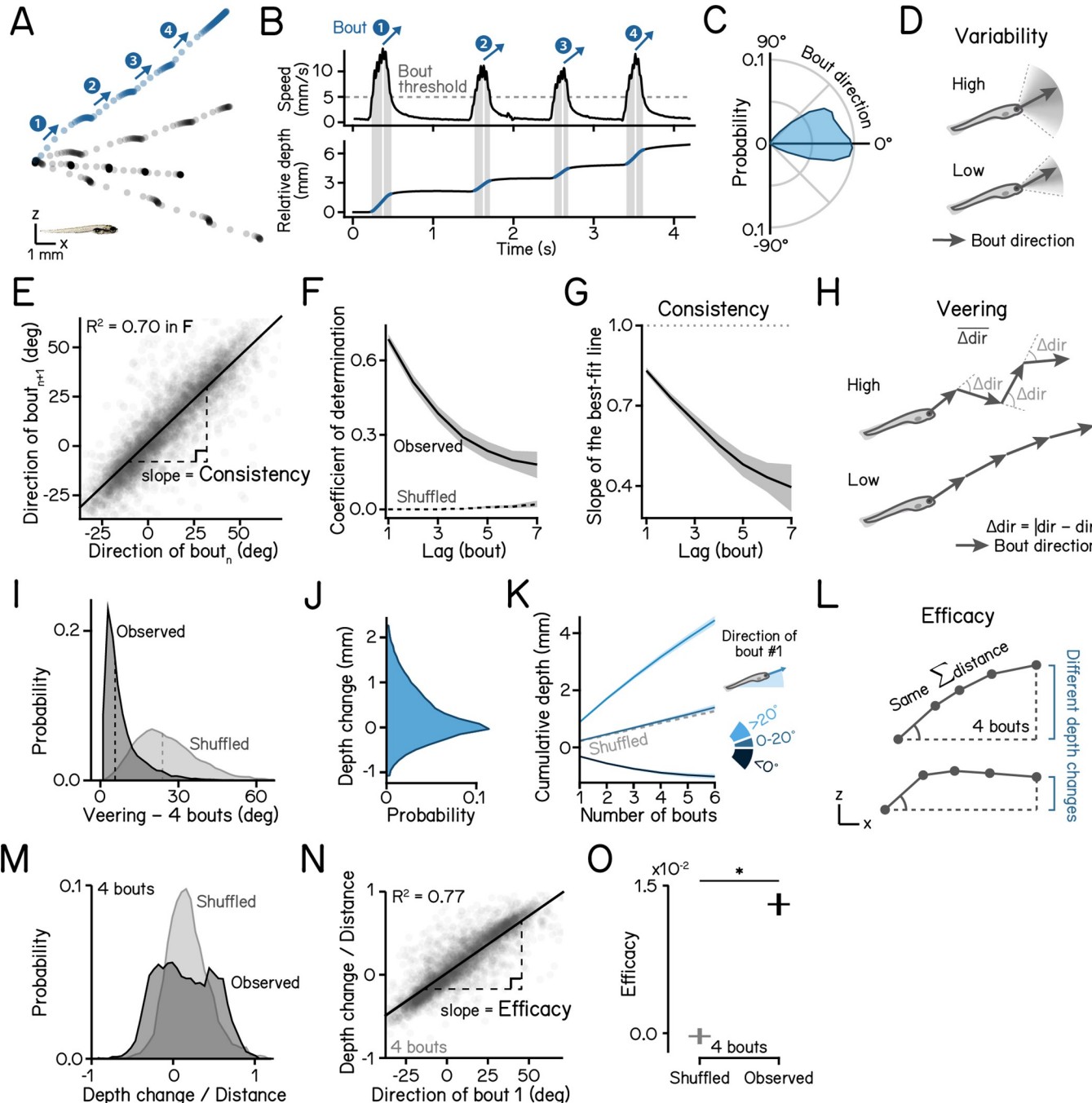

**Fig 1. Larvae navigate depth in series of consecutive bouts with consistent heading.** (A) Sample swim trajectories of 7 dpf larvae in the x/z axes. All trajectories begin from the left. Dots represent fish locations 60 ms apart (down-sampled from 166 Hz data for visualization). Arrows mark swim directions of individual bouts in the blue trajectory. Scale bar: 1 mm. (B) Time series data of the blue trajectory in A. Horizontal dashed line in the upper panel indicates the 5 mm/s threshold for bout detection. Vertical lines label the time of peak speed for each bout. Lower panel plots directions of movement (black) and body posture in the pitch axis (orange). (C) Polarized histograms (frequency polygons) of bout directions of WT zebrafish. $n$ = 123,849 bouts from 537 fish over 22 experimental repeats. (D) Schematic illustrations of bout direction variability. A wide distribution of bout directions indicates high variability. (E) Directions of the following bout plotted as a function of the current plot. Correlation coefficient is plotted in (F). Slope of the best fitted line is plotted in (G). $n$ = 63,681 bout pairs from 537 fish over 22 experimental repeats. (F) Serial correlation (autocorrelation) of swim directions across observed consecutive bouts and shuffled bouts. $N$ = 22 experimental repeats. 95% correlation confidence intervals are shown as shaded error bands. (G) Slope of the best fit line of swim directions of bout(n+lag) vs. bout(n) is defined as the swim direction consistency. $N$ = 22 experimental repeats. 95% confidence intervals of the estimated slope are shown as shaded bands. (H) Veering is quantified as the absolute change of swim directions between adjacent bouts, averaged through a bout series. A bout sequence with greater direction changes results in higher veering. (I) Veering across 4 consecutive bouts (observed) and shuffled bouts are plotted as histograms. Median

values are shown as dashed lines. $n$ = 17,155 sets of 4 bouts. $P_{median-test}$ < 1e-16. (J) Distribution of depth changes, defined as the displacement on the z axis, of a single swim bout. $n$ = 123,849 bouts from 537 fish over 22 repeats. (K) Cumulative depth changes through 4 consecutive bouts, separated by the swim direction of the first bout. (L) Schematic illustration of "depth change efficacy." A trajectory with higher efficacy achieves a greater depth change (displacement on the z axis) with the same swim distance and starting direction. (M) Histograms of depth change divided by distance across 4 swim bouts. Values of −1 or 1 represent trajectories pointing straight down or up, respectively, on the vertical axis. (N) Depth change/swim distance ratio during series of 4 bouts plotted as a function of the swim direction of the first bout. Depth change efficacy is defined as the slope of best fit line. $n$ = 17,155 sets of 4 bouts. (O) Depth change efficacy of observed and shuffled bouts. Errors indicate standard deviations. $N$ = 22 experimental repeats. $P_{t-test}$ = 8.65e-37. See also Table 1 for parameter definitions and statistics. All code and data can be found at DOI: 10.17605/OSF.IO/AER9F.

zebrafish maintain stable swim directions through a series. Consequentially, a bout series results in cumulative changes in depth (Fig 1K). In addition to swim directions, swim distances also affect depth displacement. A trajectory that allows effective depth change should allocate more displacement to the vertical axis (Fig 1L). We quantified the ratio of displacement in depth versus Euclidean swim distance and found that observed bout sequences showed a wider distribution (Fig 1M) and was highly correlated with swim directions (Fig 1N). To quantify how effectively larvae change depth, we defined depth change efficacy as the best fitted line between depth/distance ratio and the direction of the first bout in the sequence (Fig 1N). Compared to shuffled bouts, fish show significantly higher efficacy (Fig 1O; 1.31e-2 ± 1.09e-3 versus 1.37e-4 ± 7.89e-4, observed versus shuffled, mean ± SD, $P_{t-test}$ = 8.65e-37), indicating that,

**Table 1. Parameters definitions and statistics.** Refer to Fig 1.

| Parameter | Definition | Format | Value | Unit |
|---|---|---|---|---|
| Swim bout | An epoch during which fish swims faster than 5 mm/s | – | – | – |
| Swim direction | Swim direction at the time of peak speed | Median [IQR] | 9.06 [33.93] | degree |
| Direction variability | Median absolute deviation (MAD) of all swim bout directions | MAD | 16.86 | degree |
| Veering | Absolute differences of directions between adjacent bouts averaged across a series of 4 consecutive bouts | Median [IQR] | 5.61 [6.85] | degree |
| Depth change | Displacement on the vertical axis during a swim bout | Median [IQR] | 0.14 [0.76] | mm |
| Abs. depth change | Distance on the vertical axis during a swim bout | Median [IQR] | 0.34 [0.57] | mm |
| Cumu. depth change | Cumulative displacement in depth through 4 consecutive bouts | Median [IQR] | 0.49 [2.67] | mm |
| Depth change efficacy | Slope of the best fitted line of the ratio of depth change/swim distance vs. direction of the first bout in the sequence | Mean [SD] | 1.31e-2 [1.09e-3] | – |
| Consistency lag = 1 | Slope of the best fit line of direction of bout(n+lag) vs. that of bout(n) | Mean [SD] | 0.83 [2.74e-2] | – |
| lag = 2 | – | Mean [SD] | 0.73 [4.04e-2] | – |
| lag = 3 | – | Mean [SD] | 0.64 [5.66e-2] | – |
| lag = 4 | – | Mean [SD] | 0.55 [7.58e-2] | – |
| lag = 5 | – | Mean [SD] | 0.48 [0.10] | – |
| lag = 6 | – | Mean [SD] | 0.43 [0.13] | – |
| lag = 7 | – | Mean [SD] | 0.40 [0.20] | – |
| Steering gain | Slope of the best fitted line of direction vs. posture at time of the peak speed | Mean [SEM] | 0.68 [7.12e-3] | – |
| Lifting gain | Slope of the best fitted line of estimated lift vs. bout depth change | Mean [SEM] | 0.35 [8.53e-3] | – |
| Righting gain | Absolute value of the slope of the best fitted line of rotation during deceleration vs. initial posture | Mean [SEM] | 0.18 [4.04e-3] | – |
| Inter-bout interval | Duration between swim bouts when fish are inactive | Mean [SEM] | 1.83 [0.12] | s |

Definitions of navigation parameters and values of wild-type 7-day larvae. Bout parameters are reported as median across all swim bouts ($n$ = 123,849 bouts) or across 4 consecutive bouts ($n$ = 17,155). Navigation parameters and locomotion kinematics are reported as mean across experimental repeats ($N$ = 22). Refer to Methods for more information on data analysis and statistics.

given the swim direction of a bout, fish exhibit effective depth change following consecutive bouts in the sequence.

We conclude that larvae change depth effectively by performing a series of swim bouts with consistent heading. The parameters of consistency, veering, and efficacy define their ability to navigate in depth.

## Loss of gravity sensation disrupts vertical navigation

To understand whether the sensation of gravity contributes to navigation in depth, we examined behavior of 7 to 9 dpf gravity-blind larvae as they swam in complete darkness. The *otogelin* mutant fails to develop a utricular otolith (Fig 2A, arrowhead) until approximately 14 dpf [50], leaving larvae unable to sense gravity [29,30,51]. Compared to heterozygous siblings, *otog-/-* larvae showed more variable swim directions across individual bouts (Fig 2B; 20.45 ± 0.18˚ versus 21.36 ± 0.25˚, heterozygous controls versus mutants, mean ± SD, $P_{bootstrap}$ = 0.003; Table 2). In addition, series of bouts by mutants exhibited lower direction consistency (Fig 2C and Table 2), and veered more (Fig 2D; 5.50 [6.71]˚ versus 5.92 [6.84]˚, median with IQR, $P_{median\text{-}test}$ = 0.019, see also Table 2). Consequentially, gravity-blind fish were dramatically less effective at navigating in depth (Fig 2E; 1.39e-2 ± 1.41e-4 versus 1.28e-2 ± 8.86e-5, $P_{bootstrap}$

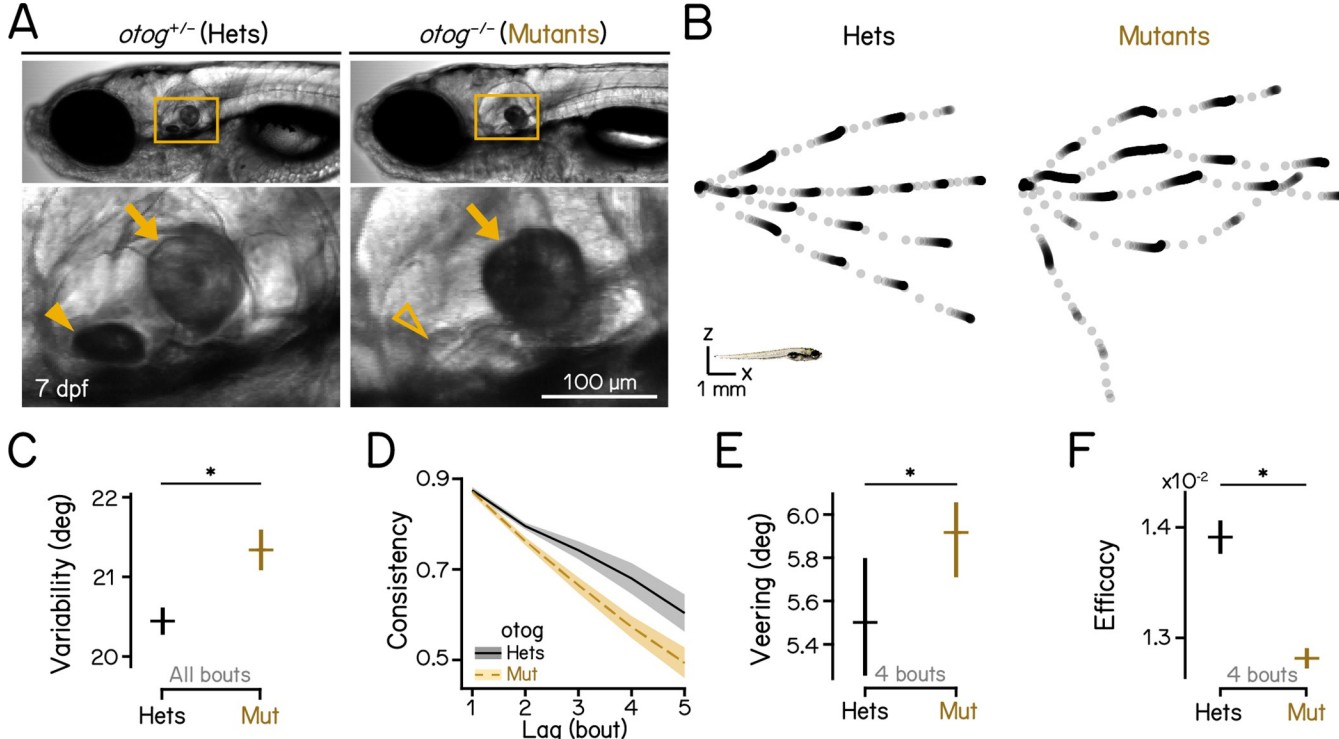

**Fig 2. Gravity-blind mutant fish have impaired vertical navigation.** (A) Homozygous *otog* mutants have intact saccular otoliths (arrows) while they lack the utricular otoliths (arrowheads) at 7 dpf. Scale bar: 100 μm. (B) Sample swim trajectories of *otog* mutants and heterozygous controls. All trajectories begin from the left. Dots represent fish locations 60 ms apart. Scale bar: 1 mm. (C) Swim direction variability, quantified as the MAD of swim directions of all bouts, compared between *otog* mutants and heterozygous controls. The means of bootstrapped MADs are plotted with error bars showing standard deviations. *n* = 14,590/10,645 bouts from 99/136 fish over 5 repeats for controls/mutants. $P_{bootstrap}$ = 3.50e-3. (D) Swim direction consistency, as defined in Fig 1G, plotted as a function of the number of bouts in the sequence. Shaded bands indicate standard deviations of the slope estimated using bootstrapping. (E) Veering through 4 consecutive bouts, as defined in Fig 1H, compared between *otog* mutants and heterozygous controls. Medians with 95% confidence intervals are plotted. *n* = 1,669/3,238 4-bout series from 99/136 fish over 5 repeats for controls/mutants. $P_{median\text{-}test}$ = 1.93e-2. (F) Depth change efficacy, as defined in Fig 1L. The means of bootstrapped slopes are plotted with error bars showing standard deviations. $P_{bootstrap}$ = 9.81e-11. See also Tables 2 and 3 for statistics. All code and data can be found at DOI: 10.17605/OSF.IO/AER9F.

**Table 2. Effects of vestibular impairments on navigation parameters.** Refer to Figs 2–4 and S3.

| Parameter | Format | Control value | Condition value | P value | Notes |
|---|---|---|---|---|---|
| *otog* mutation 1,669/3,238 4-bout sequences from 99/136 fish for hets/mutants over 5 repeats | | | | | |
| Veering (˚) | Median [95 CI] | 5.50 [5.27–5.79] | 5.92 [5.71–6.07] | $P_{median-test}$ = 1.93e-2 | 4 bouts |
| Efficacy | Mean [SD] | 1.39e-2 [1.41e-4] | 1.28e-2 [8.86e-5] | $P_{bootstrap}$ = 9.81e-11 | 4 bouts, Bootstrapped |
| Consistency lag = 1 | Mean [SD] | 0.87 [6.90e-3] | 0.87 [5.39e-3] | $P_{bootstrap}$ = 0.844 | n = 6,668/7,115 |
| lag = 2 | Mean [SD] | 0.79 [1.27e-2] | 0.76 [8.02e-3] | $P_{bootstrap}$ = 2.27e-2 | n = 3,304/4,816 |
| lag = 3 | Mean [SD] | 0.73 [1.79e-2] | 0.67 [1.11e-2] | $P_{bootstrap}$ = 6.03e-3 | n = 1,669/3,238 |
| lag = 4 | Mean [SD] | 0.66 [2.52e-2] | 0.57 [1.86e-2] | $P_{bootstrap}$ = 1.41e-2 | n = 845/2,158 |
| lag = 5 | Mean [SD] | 0.60 [2.99e-2] | 0.50 [1.90e-2] | $P_{bootstrap}$ = 3.98e-3 | n = 443/1,399 |
| **Tangential lesions** 3,930/2,530 4-bout sequences from 40/25 fish for controls/lesions over 8 repeats | | | | | |
| Veering (˚) | Median [95 CI] | 6.44 [6.24–6.59] | 7.27 [6.91–7.50] | $P_{median-test}$ = 1.31e-8 | 4 bouts |
| Efficacy | Mean [SD] | 1.39e-2 [1.06e-4] | 1.26e-2 [1.27e-3] | $P_{bootstrap}$ = 5.19e-16 | 4 bouts, Bootstrapped |
| Consistency lag = 1 | Mean [SD] | 0.88 [5.75e-3] | 0.86 [6.23e-3] | $P_{bootstrap}$ = 4.48e-2 | n = 10,866/7,022 |
| lag = 2 | Mean [SD] | 0.82 [9.10e-3] | 0.78 [8.16e-3] | $P_{bootstrap}$ = 1.48e-3 | n = 6,542/4,213 |
| lag = 3 | Mean [SD] | 0.78 [1.26e-2] | 0.72 [1.62] | $P_{bootstrap}$ = 1.07e-3 | n = 3,930/2,530 |
| lag = 4 | Mean [SD] | 0.76 [1.00e-2] | 0.69 [1.87] | $P_{bootstrap}$ = 2.03e-2 | n = 2,342/1,514 |
| lag = 5 | Mean [SD] | 0.76 [2.05e-2] | 0.66 [2.37] | $P_{bootstrap}$ = 1.03e-2 | n = 1,384/892 |
| **Vestibulospinal lesions** 3,554/2,920 4-bout sequences from 79/97 fish for controls/lesions over 8 repeats | | | | | |
| Veering (˚) | Median [95 CI] | 9.35 [9.06–9.71] | 9.79 [9.39–10.11] | $P_{median-test}$ = 0.154 | 4 bouts |
| Efficacy | Mean [SD] | 9.78e-3 [1.80e-4] | 1.19e-2 [1.19e-4] | $P_{bootstrap}$ = 1.61e-24 | 4 bouts, Bootstrapped |
| Consistency lag = 1 | Mean [SD] | 0.70 [4.70e-3] | 0.82 [4.73e-3] | $P_{bootstrap}$ = 2.49e-23 | n = 9,976/9,457 |
| lag = 2 | Mean [SD] | 0.60 [1.23e-2] | 0.74 [8.95e-3] | $P_{bootstrap}$ = 2.31e-15 | n = 5,765/5,073 |
| lag = 3 | Mean [SD] | 0.54 [1.86e-2] | 0.70 [1.26e-2] | $P_{bootstrap}$ = 1.01e-14 | n = 3,554/2,920 |
| lag = 4 | Mean [SD] | 0.49 [2.39e-2] | 0.66 [2.49e-2] | $P_{bootstrap}$ = 3.80e-9 | n = 2,284/1,782 |
| lag = 5 | Mean [SD] | 0.45 [2.40e-2] | 0.64 [2.85e-2] | $P_{bootstrap}$ = 8.28e-8 | n = 1,506/1,122 |
| **INC/nMLF lesions** 4,355/4,743 4-bout sequences from 30/31 fish for controls/lesions over 6 repeats | | | | | |
| Veering (˚) | Median [95 CI] | 2.57 [5.90–6.21] | 3.23 [6.40–6.82] | $P_{median-test}$ = 2.13e-3 | 4 bouts |
| Efficacy | Mean [SD] | 1.34e-2 [9.61e-5] | 1.23e-2 [1.16e-4] | $P_{bootstrap}$ = 4.28e-11 | 4 bouts, Bootstrapped |
| Consistency lag = 1 | Mean [SD] | 0.82 [5.44e-3] | 0.80 [6.50e-3] | $P_{bootstrap}$ = 1.10e-2 | n = 12,433/1,3670 |
| lag = 2 | Mean [SD] | 0.72 [8.66e-3] | 0.68 [1.27e-2] | $P_{bootstrap}$ = 1.90e-2 | n = 7,362/8,048 |
| lag = 3 | Mean [SD] | 0.64 [1.26e-2] | 0.58 [2.06e-2] | $P_{bootstrap}$ = 1.03e-2 | n = 4,355/4,743 |
| lag = 4 | Mean [SD] | 0.57 [1.64e-2] | 0.50 [2.53e-2] | $P_{bootstrap}$ = 4.46e-3 | n = 2,520/2,772 |
| lag = 5 | Mean [SD] | 0.53 [2.53e-2] | 0.45 [2.87e-2] | $P_{bootstrap}$ = 2.90e-2 | n = 1,390/1,522 |

Number of fish, number of bouts, and methods of statistical analysis and *P* values are reported. All statistical analyses are two-tailed. Refer to Methods for more information on statistics.

= 0.001), in congruence with their high veering. We conclude that loss of gravity sensation destabilizes swim directions leading to less effective vertical navigation.

## Ablation of gravity-sensitive vestibular neurons disrupts vertical navigation and swim kinematics

Previous studies demonstrate that gravity-sensitive ascending neurons of the tangential vestibular nucleus [52] and descending vestibulospinal neurons of the lateral vestibular nucleus [53] encode body tilt and regulate postural behaviors [24,30,33–36]. We adopted a loss-of-function approach, using a pulsed infrared laser to ablate genetically defined populations of ascending neurons in the tangential nucleus (S2A and S2B Fig). In addition, we reanalyzed a data set [34] comprised of larvae with lesioned descending vestibulospinal neurons (S2 Fig).

Loss of ascending neurons in the tangential nucleus (Fig 3A and 3B) recapitulated disruption to swim directions and vertical navigation seen in gravity-blind fish. After lesions, fish had more variable swim directions (Fig 3C; 20.86 ± 0.14˚ versus 22.08 ± 0.23˚, control versus lesions, mean ± SD, $P_{bootstrap} < 0.001$. See also Table 2). Direction consistency was reduced (Fig 3D and Table 2) and veering increased (Fig 3E; 6.44 [6.11]˚ versus 7.27 [6.81]˚, median with IQR, $P_{median-test} < 0.001$), disrupting depth change efficiency (Fig 3F; 1.39e-2 ± 1.06e-4 versus 1.26e-2 ± 1.27e-3, $P_{bootstrap} < 0.001$).

Similar to loss of ascending neurons, lesions of descending vestibulospinal neurons (S1 and S3 Figs) also increased swim direction variability (S3 Fig; 19.93 ± 0.17˚ versus 24.50 ± 0.20˚, $P_{bootstrap} < 0.001$). However, vestibulospinal-lesioned larvae adopted more consistent swim directions through consecutive swim bouts (S3 Fig and Table 2) and achieved greater depth changes through sequence of bouts compared to sibling controls (S3 Fig; 9.78e-3 ± 1.80e-4 versus 1.19e-2 ± 1.19e-4, $P_{bootstrap} < 0.001$).

We conclude that ascending neurons in the tangential nucleus, and not vestibulospinal neurons, are critical to maintain stable and consistent heading required for effective navigation in depth.

Loss of vestibular function should disrupt posture and locomotor behaviors. We investigated kinematic features that determine swim directions in the vertical axis (S4 Fig). During bouts, larval zebrafish utilize a three-step strategy that allows them to climb and dive while maintaining their preferred horizontal posture [22,49,54] (S4 Fig and Table 1): First, larvae steer by rotating their body. Next, they coordinate propulsive forces generated by undulatory thrust and pectoral-fin-based lift. Finally, they rotate back toward their preferred posture. The strength of each of these behaviors can be parameterized as a gain, to indicate the how strongly the fish steer (S4 Fig), achieve vertical translocation through lift (S4 Fig), and restore posture (S4 Fig).

Compared to sibling controls, vestibular-impaired larvae exhibited higher steering gain (S4 Fig; $P_{t-test} < 0.001$, $P_{t-test} < 0.001$, $P_{t-test} = 0.002$; Table 2) and lower lifting gain (S4 Fig; $P_{t-test} < 0.001$, $P_{t-test} = 0.036$, $P_{t-test} = 0.005$, for *otogelin* mutants, tangential, and vestibulospinal lesions, respectively). *otogelin* mutants and vestibulospinal lesions resulted in significant decrease in the righting gain (S4 Fig; $P_{t-test} < 0.001$, $P_{t-test} = 0.180$, $P_{t-test} = 0.001$, for *otogelin* mutants, tangential, and vestibulospinal lesions, respectively).

Taken together, these data confirm that, as expected, perturbations of vestibular sensation or vestibular neurons disrupt swim kinematics. Specifically, after lesions, larvae can still change depth but they do so with more eccentric posture and poorly coordinated lift (S4 Fig). These results are consistent with the increased variability seen in *otogelin* mutants (Fig 2B), and after lesions of either ascending tangential nucleus neurons (Fig 3C) or vestibulospinal neurons (S3 Fig). The different effects of tangential and vestibulospinal lesions on swim kinematics and depth navigation suggests that navigational strategy is disassociated from the steering, lifting, and righting behaviors that define individual swim bouts.

## Gravity-guided stabilization of swim directions for vertical navigation is mediated by the midbrain nucleus INC/nMLF

Ascending neurons of the tangential nucleus project to the INC/nMLF, which sends descending axons to the spinal cord to control locomotion [24,55–58] (Fig 4A). We reasoned that the INC/nMLF might be the final supraspinal node in a circuit for gravitational control of heading during vertical navigation. We therefore lesioned large descending neurons in the INC/nMLF (Figs 4A, 4B, and S2). Ablation slightly decreased directional variability (Fig 4C; 15.08 ± 0.13˚ versus 14.18 ± 0.12˚, controls versus lesions, $P_{bootstrap} < 0.001$). Similar to effects on navigation

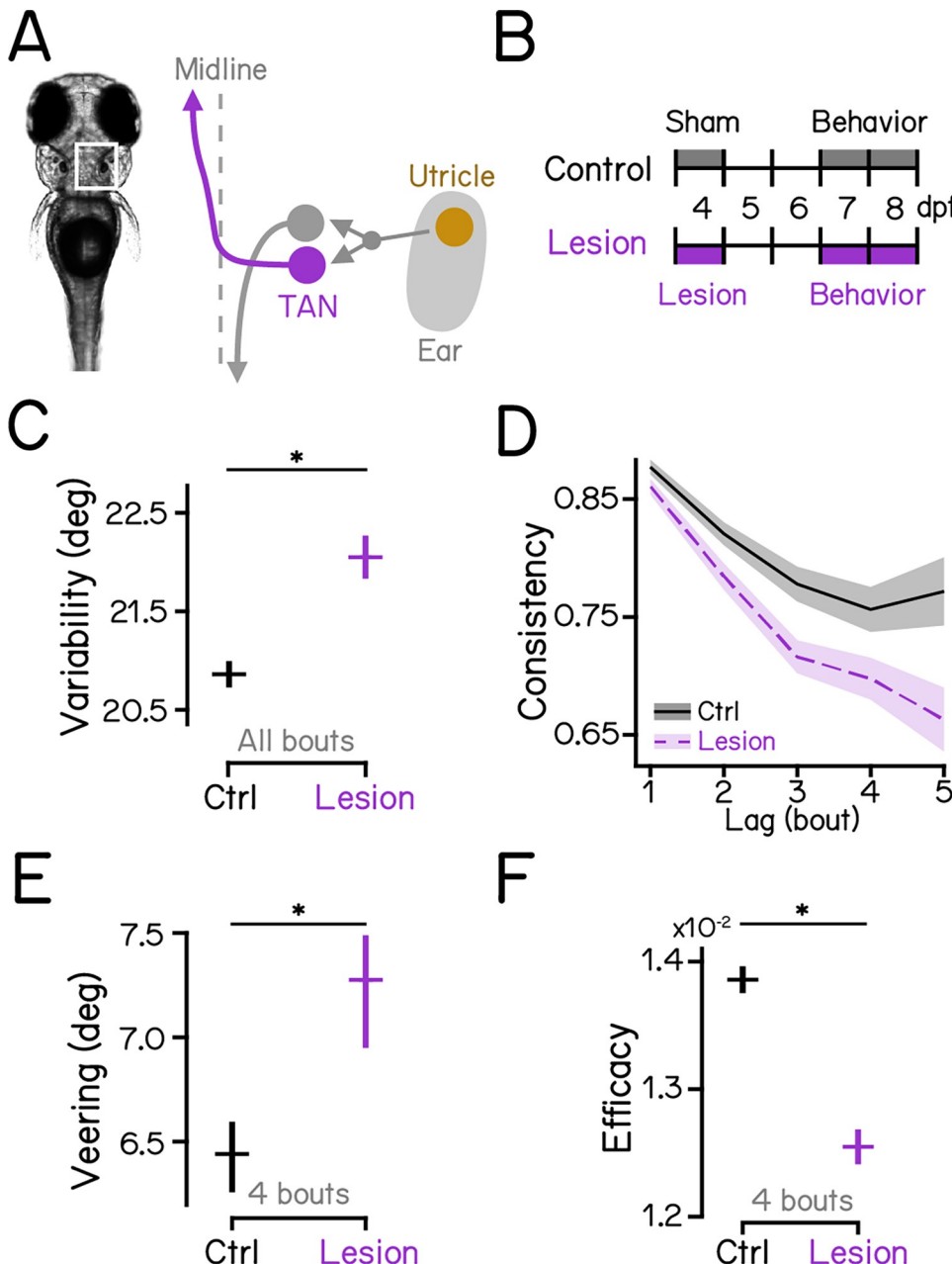

**Fig 3. Ascending neurons in the tangential nucleus are indispensable for vertical navigation.** (A) Schematic view of the inner-ear utricular otolith and the vestibular pathways in the hindbrain of zebrafish. Utricle: utricular otoliths (yellow); TAN: the tangential vestibular nucleus (magenta). (B) Diagrams of experimental procedures for lesions of the tangential nucleus and behavioral assays. See S2 Fig for examples of lesions. (C) Swim direction variability compared between tangential-lesioned larvae and controls. The means of bootstrapped MADs are plotted with error bars indicating standard deviations. $n$ = 17,797/11,417 bouts from 40/25 fish over 8 repeats for controls/lesions. $P_{bootstrap}$ = 6.95e-6. (D) Swim direction consistency plotted as a function of the number of bouts in the sequence. Shaded bands indicate standard deviations of the slope estimated using bootstrapping. (E) Veering through 4 consecutive bouts plotted in median with 95% confidence intervals. $n$ = 3,930/2,530 4-bout series from 40/25 fish over 8 repeats for controls/lesions. $P_{median-test}$ = 1.31e-8. (F) Depth change efficacy plotted as bootstrapped means with error bars showing standard deviations. $P_{bootstrap}$ = 5.19e-16. See also Tables 2 and 3 for statistics. All code and data can be found at DOI: 10.17605/OSF.IO/AER9F.

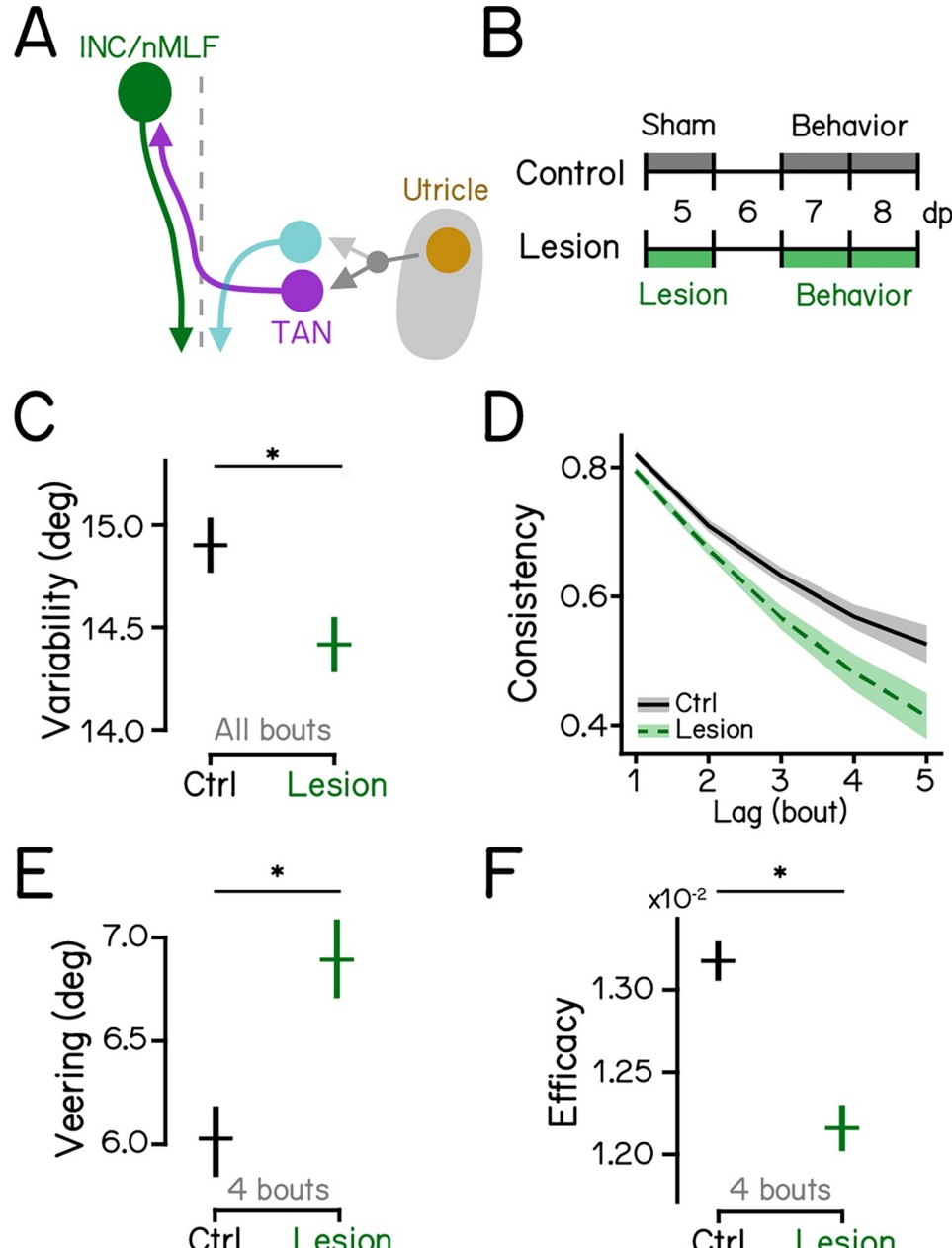

**Fig 4. Descending neurons in the INC/nMLF are indispensable for vertical navigation.** (A) Schematic diagram of the hindbrain-midbrain circuit. Descending neurons in the INC/nMLF (green) receive contralateral vestibular inputs from the tangential nucleus (magenta). (B) Experimental diagram of INC/nMLF lesions and behavior assays. Refer to S2 Fig for INC/nMLF lesions. (C) Swim direction variability compared between INC/nMLF-lesioned larvae and controls with Bootstrapped standard deviations shown as error bars. $n$ = 21,583/23,699 bouts from 30/31 fish over 6 repeats for controls/lesions. $P_{bootstrap}$ = 1.90e-7. (D) Swim direction consistency plotted as a function of the number of bouts in the sequence. Shaded bands indicate standard deviations of the slope estimated by bootstrapping. (E) Veering through 4 consecutive bouts plotted in median with 95% confidence intervals. $n$ = 4,355/4,743 4-bout series from 30/31 fish over 6 repeats for controls/lesions. $P_{median-test}$ = 2.13e-3. (F) Depth change efficacy plotted as bootstrapped means with error bars showing standard deviations. $P_{bootstrap}$ = 4.28e-11. TAN, the tangential nucleus; INC, interstitial nucleus of Cajal; nMLF, nucleus of the medial longitudinal fasciculus. See also Tables 2 and 3 for statistics. All code and data can be found at DOI: 10.17605/OSF.IO/AER9F.

seen in *otogelin* mutants and after ascending tangential neuron lesions, larvae with INC/nMLF lesions showed reduced consistency of swim directions (Fig 4D and Table 2), increased veering (Fig 4E; 2.57 [3.86]° versus 3.23 [4.53]°, median with IQR, $P_{median\text{-}test}$ = 0.002), and were less effective at changing depth (Fig 4F; 1.34e-2 ± 9.61e-5 versus 1.23e-2 ± 1.16e-4, $P_{bootstrap} <$ 0.001).

Larvae with lesions of descending neurons in the INC/nMLF recapitulated phenotypes observed in *otogelin* mutants and after lesions of ascending neurons in the tangential vestibular nucleus. Specifically, all 3 disrupt heading consistency across a series of bouts and show increased veering. Together, these decrease the efficacy of changing depth. Taken together, our results reveal a circuit from the inner ear to the spinal cord responsible for gravitational control of heading during vertical navigation.

## Discussion

We define a circuit that uses gravitational information to control heading for effective vertical navigation. Larvae use a series of swim bouts with consistent heading to change depth in the dark. Loss of either the utricular otoliths, utricle-recipient ascending neurons in the tangential vestibular nucleus, or spinal projecting neurons in the INC/nMLF all caused fish to swim with more variable heading and excessive veering, leading to less effective depth changes. Taken together, this work reveals ancient brainstem architecture that uses gravitational cues to move effectively through the world.

### Circuit architecture and computations for vertical navigation

Our work argues that the INC/nMLF contributes to vertical navigation. In larval zebrafish, the INC/nMLF is best known for its role in regulating swim posture and speed [55–58]. Further, the tangential-INC/nMLF circuit controls a vestibular-induced body bend reflex that allows fish to maintain posture in the roll (barbecue) axis [24]. Across mammals, INC/nMLF is perhaps best known as the site of the neural integrator for vertical/torsional eye position [59–61], a computation that acts as a short-term memory for motor commands [62]. Saccades that move the eyes to a new position are instantiated with short bursts of neuronal activity. When integrated, these bursts provide the signal necessary for extraocular motor neurons to maintain muscle tension, stabilizing gaze at the new position. Similarly, integration transforms vestibular representations of head velocity into eye position for a proper vestibulo-ocular reflex [63].

We propose that the utricle-tangential-INC/nMLF circuit stores and uses a short-term memory of gravity-derived signals for vertical navigation. We observed that larval zebrafish maintain their heading across a series of bouts. The timescale of this phenomena suggests the existence of a short-term memory for commands to shape posture and kinematics. *otogelin* mutants show profoundly disrupted vertical navigation in darkness. Therefore, while heading may persist partly due to inertia, a gravity-derived neural command must contribute as well. Intriguingly, the idea that oculomotor integrator circuits might serve a role in navigation has a recent parallel: in larval zebrafish, the nucleus prepositus hypoglossi (NPH), best known as the neural integrator for horizontal eye movements, may integrate self-motion signals in the yaw plane [47].

Unlike perturbations to the utricle-tangential-INC/nMLF circuit, larvae without vestibulospinal neurons navigate more consistently and effectively. While puzzling, this finding is important for 3 reasons. First, it demonstrates that not all lesions to the utricle-recipient neurons lead to disrupted navigation. Second, it shows that loss of gravity sensation and lesions of vestibular nuclei perturb swim kinematics similarly, leading to increased variability in bout direction. Third, it suggests a dissociation between gravity-guided heading and swim

**Table 3. Effects of vestibular impairments on locomotion parameters.** Refer to Figs 2–4 and S4.

| Parameter | Format | Control value | Condition value | P value | Notes |
|---|---|---|---|---|---|
| *otog* mutation 14,590/10,645 bouts from 99/136 fish for hets/mutants over 5 repeats | | | | | |
| Variability (°) | Mean [SD] | 20.45 [0.18] | 21.36 [0.25] | $P_{bootstrap}$ = 3.50e-3 | Bootstrapped |
| Steering Gain | Mean [SEM] | 0.69 [1.01e-2] | 0.85 [1.52e-2] | $P_{t-test}$ = 2.33e-5 | 5 repeats |
| Lifting Gain | Mean [SEM] | 0.32 [1.45e-2] | 0.15 [2.95e-2] | $P_{t-test}$ = 7.09e-4 | 5 repeats |
| Righting Gain | Mean [SEM] | 0.17 [1.08e-2] | 0.09 [1.03e-2] | $P_{t-test}$ = 6.76e-4 | 5 repeats |
| Inter-bout interval (s) | Mean [SEM] | 1.69 [0.25] | 1.19 [0.30] | $P_{t-test}$ = 0.23 | 5 repeats |
| **Tangential lesions** 17,797/11,417 bouts from 40/25 fish for controls/lesions over 8 repeats | | | | | |
| Variability (°) | Mean [SD] | 20.86 [0.14] | 22.08 [0.23] | $P_{bootstrap}$ = 6.95e-6 | Bootstrapped |
| Steering Gain | Mean [SEM] | 0.70 [1.98e-2] | 0.80 [2.23e-2] | $P_{t-test}$ = 7.41e-3 | 8 repeats |
| Lifting Gain | Mean [SEM] | 0.31 [2.88e-2] | 0.20 [4.02e-1] | $P_{t-test}$ = 3.57e-2 | 8 repeats |
| Righting Gain | Mean [SEM] | 0.14 [6.59e-3] | 0.13 [9.73e-3] | $P_{t-test}$ = 0.180 | 8 repeats |
| Inter-bout interval (s) | Mean [SEM] | 1.36 [0.12] | 1.35 [0.13] | $P_{t-test}$ = 0.96 | 8 repeats |
| **Vestibulospinal lesions** 18,366/18,363 bouts from 79/97 fish for controls/lesions over 8 repeats | | | | | |
| Variability (°) | Mean [SD] | 19.93 [0.17] | 24.50 [0.20] | $P_{bootstrap}$ = 1.42e-60 | Bootstrapped |
| Steering Gain | Mean [SEM] | 0.60 [2.27e-2] | 0.75 [3.40e-2] | $P_{t-test}$ = 2.32e-3 | 8 repeats |
| Lifting Gain | Mean [SEM] | 0.41 [2.74e-2] | 0.27 [3.16e-2] | $P_{t-test}$ = 5.90e-3 | 8 repeats |
| Righting Gain | Mean [SEM] | 0.17 [5.38e-3] | 0.11 [1.22e-2] | $P_{t-test}$ = 1.00e-3 | 8 repeats |
| Inter-bout interval (s) | Mean [SEM] | 1.97 [0.22] | 1.40 [0.08] | $P_{t-test}$ = 3.01e-2 | 8 repeats |
| **INC/nMLF lesions** 21,583/23,699 bouts from 30/31 fish for controls/lesions over 6 repeats | | | | | |
| Variability (°) | Mean [SD] | 15.08 [0.13] | 14.18 [0.12] | $P_{bootstrap}$ = 1.90e-7 | Bootstrapped |
| Steering Gain | Mean [SEM] | 0.69 [1.31e-2] | 0.71 [1.92e-2] | $P_{t-test}$ = 0.335 | 6 repeats |
| Lifting Gain | Mean [SEM] | 0.34 [1.65e-2] | 0.34 [1.71e-2] | $P_{t-test}$ = 0.866 | 6 repeats |
| Righting Gain | Mean [SEM] | 0.20 [9.50e-3] | 0.19 [9.30e-3] | $P_{t-test}$ = 0.394 | 6 repeats |
| Inter-bout interval (s) | Mean [SEM] | 2.72 [0.28] | 3.38 [0.20] | $P_{t-test}$ = 8.53e-2 | 6 repeats |

Number of fish, number of bouts, and methods of statistical analysis and P values are reported. All statistical analyses are two-tailed. Refer to Methods for more information.

kinematics that describe individual swim bouts [34]. Future work with more comprehensive kinematic modeling will elucidate how individual movements are connected and grouped for complex navigation behavior.

Why might fish without vestibulospinal neurons veer less? Fish with vestibulospinal lesions had shorter inter-bout intervals (Table 3). During the inter-bout interval, larvae experience nose-down torques due to their top-heavy body composition [27]. A shorter interval would reduce the effect of this nose-down acceleration, and orientation from bout to bout would be more stable. Alternatively, or additionally, vestibulospinal input to the spinal circuits, which underlies postural-driven swim initiation [34], might increase veering. Removal of vestibulospinal input in fish with intact utricle-tangential-INC/nMLF circuit would lead to more stable swim directions from bout to bout. Defining and comparing the spinal targets of vestibulospinal neurons [64] to the targets of descending neurons in the INC/nMLF [57,65] would speak to this proposal. Our findings set the stage to explore the integration of gravity-derived information from direct and indirect brainstem projections to the spinal cord.

## Are larval zebrafish truly navigating depth?

Environmental cues such as light, food, or behavioral state can guide vertical aquatic navigation. Many species of aquatic animals migrate up/down [66], following the 24-h cycle of the

zooplankton diel vertical migration [67]. Ocean sunfish will perform deep dives during the day to feed in the mesopelagic zone, returning to the surface to warm up [68–70]. Elephant seals dive during sleep to avoid predators [71]. Most pertinently, larval zebrafish can dive/surface following changes to illumination [42–44] and anxiogenic/anxiolytic drugs [72], and tend to occupy the top third of the water column in a tall (36 cm) tank [73].

All behavior in the current study was measured in complete darkness without a defined goal, raising questions of terminology. One proposal would classify the behavior we observe as "gravity-guided orientation," because larvae seek to arrive at a more preferable depth, rather than navigating to a specific location like a nest [74,75]. However, "orientation" may refer to both a stationary/perceptual response and a locomotor activity in its definition [76]. To avoid ambiguity, we refer to the behavior we see as "gravity-guided navigation," consistent with a broader view of what comprises navigation [48]. By studying unconstrained vertical navigation, our work sets the foundation for exploration of more complex behavior paradigms. For example, future work should introduce perturbations, enable goal-directed tasks, and deliver additional stimuli to understand gravity's influence on navigation.

## Limitations

A potential caveat of our loss-of-function approach is that the lesions were done at different ages. We did not observe regrowth of neuronal cell bodies at any of the lesion sites by the time of behavior assays among the fish we examined, so we do not expect differential regeneration to affect our results. However, larvae might exhibit different levels of adaptation to the impairments before behavioral assessment. Given the consistency in effects across constitutive loss (mutants), 4 dpf (tangential) and 5 dpf (INC/nMLF) lesions, we think effects of adaptation are likely minimal. Future work with longitudinal behavioral assays will permit investigation of the mechanisms of adaptation and rehabilitation after circuit disruption. Finally, the field of view of our SAMPL apparatus (20 mm × 20 mm [49]) limits the amount of long trajectories we can collect. As we looked further down the bout sequence, we record drastically fewer bouts (S1 Fig). Future work using apparatus with a larger field of view will allow analysis of longer trajectories.

We evaluated the effects of manipulations by comparing perturbed fish and control siblings. Notably, we observed large variations among navigation parameters across control fish from different data sets. We have previously reported that kinematic parameters vary systematically among wild-type larvae from different genetic backgrounds [49]. Given that animals used in different experiments in this study are unrelated in genotypes and backgrounds, we infer that the variations arise from genetic backgrounds. Our work underscores the utility of controlling genetic background to allow direct comparison across conditions.

## Conclusion

We define a sensorimotor circuit that uses evolutionarily conserved brainstem architecture to transform gravitational signals into stable heading for effective vertical navigation. The work lays a circuit-level foundation to understand persistent signals that guide locomotion and how vestibular inputs allow animals to move effectively through their environment.

## Materials and methods

### Ethics statement

All procedures involving larval zebrafish (*Danio rerio*) were approved by the NYU Grossman School of Medicine Institutional Animal Care & Use Committee (IACUC) under approval

number IA16-00561. Zebrafish embryos and larvae were raised at 28.5°C on a standard 14:10 h light:dark cycle with the lights on from 9 AM to 11 PM. Larvae were raised at a density of 20 to 50 in 25 to 40 ml of E3 medium in 10 cm petri dishes before 5 dpf. After 5 dpf, larvae were maintained at densities under 30 larvae per 10 cm petri dish and were fed cultured rotifers (Reed Mariculture) daily.

### Fish lines

Experiments were done using wild-type fish with a mixed background of AB, TU, WIK, and SAT. Larvae for lesion experiments were on the *mitfa*$^{-/-}$ background to remove pigment. Photoablations of ascending neurons in the tangential nucleus were performed in *Tg(-6.7Tru. Hcrtr2:Gal4-VP16)*; *Tg(UAS:EGFP)* [35]. Photoablations of vestibulospinal neurons and neurons of the INC/nMLF were performed on the *Is(nefma:hsp70l-LOXP-Gal4FF)*$^{stl601Tg}$ [77]; *Tg (UAS:EGFP)* background, henceforth called *Tg(nefma::EGFP)*. *otogelin* mutants were *rock solo otogelin* knockouts$^{vo66/vo66}$ [50].

### Vestibular manipulations and photoablations

*otogelin* mutants were screened at 2 dpf for bilateral loss of utricular otoliths. Photoablations of ascending neurons of the tangential nucleus were performed in *Tg(-6.7Tru.Hcrtr2:Gal4-VP16; UAS:EGFP)* larvae at 4 dpf. Lesions of the vestibulospinal neurons and the INC/nMLF were performed in *Tg(nefma::EGFP)* on days 6–7 and 5 dpf, respectively.

All lesions were done using a 2-photon laser as previously described [34]. Briefly, larvae were anesthetized in 0.2 mg/ml MESAB and then mounted in 2% low-melting point agarose. Neurons of interest were identified and imaged using an upright microscope (ThorLabs Bergamo) with an 80 MHz Ti:Sapphire oscillator-based laser at 920 nm (SpectraPhysics MaiTai HP). A separate high-power pulsed infrared laser (SpectraPhysics Spirit 8W) was used for photoablation (1,040 nm, 200 kHz repetition rate, 400 fs pulse duration, 1–4 pulses per neuron over 10 ms at 25–75 nJ per pulse). Lesion controls were sibling fish and were anesthetized for comparable durations to lesioned larvae. Lesioned and control sibling larvae were allowed to recover at 28.5°C until behavioral measurements.

### Behavioral measurements

Methods to measure behavior, including apparatus design, hardware, software, and procedures, have been extensively detailed [49]. Briefly, larvae at 7 dpf were transferred from petri dishes to behavior chambers. To achieve the necessary spatial and temporal resolution, our behavioral apparatus has a limited field of view and only measures a subset of bouts made by an individual fish. We load each apparatus with multiple fish simultaneously to collect enough bouts to estimate navigation parameters. A given experiment consists of all bouts produced by siblings from a single clutch that were run on a given day. Behavioral recordings were started on day 7 between 9 AM and noon, and lasted for approximately 48 h in complete darkness. After 24 h of recording, programs were paused for 30 min for feeding; 1 to 2 ml of rotifer culture was added to each chamber. Larvae were removed from the apparatus approximately 48 h after the start of the experiment. Data during the circadian day were used for all analyses.

### Behavioral analysis

Behavior data were analyzed using our previously published pipeline [49]. In brief, the location of a fish and its pitch axis posture were extracted and saved in real time when a single fish was present in the field of view with its body plane perpendicular to the light path. Swim

parameters during swim bouts (defined as a duration where fish swam faster than 5 mm/s) were extracted. Intervals between swim bouts where speed was lower than 5 mm/s were considered inter-bout intervals. Swim bouts were aligned at the time of the peak speed for subsequent analysis. Only consecutive swim bouts were used for autocorrelation analysis, which examined the relationship between bouts with different lags. The lag between 2 bouts in the same bout series was determined by the number of inter-bout intervals that elapsed in-between. A lag of 1 defines adjacent swim bouts. Bout pairs with different lags were extracted from sequential bouts in a series. For example, a series of 4 consecutive bouts yields 3 pairs of adjacent bouts, 2 pairs of lag-2 bouts, and 1 pair of lag-3 bouts. Table 1 defines each analysis parameter.

## Statistics

All measurements and statistics have been reported in Tables 1 to 3, including expected value, variance, and confidence intervals of parameters. Sample sizes (e.g., number of experimental repeats, number of fish, bout numbers, and number of bout sequences) are reported in figure legends.

For wild-type data, each experimental repeat (i.e., biological replicate) consists of all bouts produced by siblings from a single clutch that were run on a given day. Different clutches of animals, on different days, were treated as independent. As wild-type data is straightforward to collect and abundant, each repeat consists of sufficient bouts to stably estimate each navigation parameter. The "$N$" used for statistical analysis was the number of repeats (Table 1).

For manipulations, a repeat similarly consists of siblings run as "control" and "perturbed" on a given day. Lesioning a single fish takes considerable time, and so a single repeat for these experiments yielded an order of magnitude fewer bouts than a wild-type repeat. To estimate consistency and efficiency, we pooled data across repeats and used a permutation test to explicitly compute a $p$ value ($P_{bootstrap}$). Specifically, after pooling we resampled (with replacement) the data from each condition and computed the expected value for control and perturbed data sets 100 to 500 times. We used these repeats to generated test statistics and $p$ values (Table 2). Because the $p$ value was computed explicitly and not estimated with respect to a theoretical probability distribution, we did not report an "$N$." For veering, one measurement was calculated from each set of 4-bout sequences. Because veering values are nonparametric, the median test was used to determine significance between the 2 conditions. Since locomotor parameters could be estimated from single repeats, we did so and reported the $N$ in Table 3.

## Supporting information

**S1 Fig. Vestibular contribution to swim kinematics. (A)** Coefficient of determination plotted as a function of lag, extended to 12 bouts in a series. Values are calculated using all bouts. Standard deviations of bootstrapped data shown as shaded errors. **(B)** Correlation $p$ value plotted as a function of lag. Values are calculated using all bouts. Errors indicate standard deviations of bootstrapped results. **(C)** Number of recorded bout sequences plotted as the length of the bout sequence. All code and data can be found at DOI: 10.17605/OSF.IO/AER9F. (TIF)

**S2 Fig. Example larvae before and after photoablation. (A)** Schematic diagram of vestibular circuits in the brain-stem. Utricle: utricular otoliths (yellow); TAN: the tangential vestibular nucleus (magenta); VS: vestibulospinal neurons (cyan); INC/nMLF: the interstitial nucleus of Cajal/the nucleus of the medial longitudinal fasciculus (green). **(B)** Before and after lesions of the tangential vestibular nucleus (circled) in a 4 dpf larvae. Scale bar: 50 μm. **(C)** Before and

after lesions of the vestibulospinal nucleus (circled) in a 6 dpf larvae. Scale bar: 50 μm. **(D)** Before and after lesions of large neurons in the INC/nMLF (circled) in a 5 dpf larvae. Scale bar: 50 μm. All imaging data can be found at DOI: 10.17605/OSF.IO/AER9F.
(TIF)

**S3 Fig. Lesions of vestibulospinal neurons increase postural variability but stabilize veering, improving vertical navigation. (A)** Schematic view of the inner-ear utricular otolith and the vestibular pathways in the hindbrain of zebrafish. Utricle: utricular otoliths (yellow); VS: vestibulospinal neurons (cyan). **(B)** Diagrams of experimental procedures for lesions of the vestibulospinal nucleus and behavioral assays. See S2 Fig for examples of lesions. **(C)** Swim direction variability compared between vestibulospinal-lesioned larvae and controls. The means of boot-strapped MADs are plotted with error bars showing standard deviations. $n$ = 18,366/ 18,363 bouts from 79/97 fish over 8 repeats for controls/lesions. $P_{bootstrap}$ = 1.42e-60. **(D)** Swim direction consistency plotted as a function of the number of bouts in the sequence. Shaded bands indicate standard deviations of the slope estimated using bootstrapping. **(E)** Veering through 4 consecutive bouts plotted in median with 95% confidence intervals. $n$ = 3,554/2,920 4-bout series from 79/97 fish over 8 repeats for controls/lesions. $P_{median-test}$ = 0.154. **(F)** Depth change efficacy plotted as bootstrapped means with error bars showing standard deviations. $P_{bootstrap}$ = 1.61e-24. See also Tables 2 and 3 for statistics. All code and data can be found at DOI: 10.17605/OSF.IO/AER9F.
(TIF)

**S4 Fig. Vestibular contribution to swim kinematics. (A)** Schematic diagram showing steering, lifting, and righting during a swim bout. Larvae steer toward targeted direction during acceleration (red arrow), use pectoral fins to assist in depth changes (blue), and restore posture to horizontal during deceleration (green arrow). Z displacement generated by lifting (blue) is estimated by subtracting theoretical displacement in depth, calculated from the head direction and x distance, from the total depth change. **(B)** Steering gain is defined as the slope of the best fit line of posture at the time of the peak speed vs. swim direction. $n$ = 121,979 bouts from 537 fish. **(C)** Lift gain is defined as the slope of the best fit line of estimated lift vs. depth change of the swim bout. Pectoral-fin amputation reduces lift (dashed line). $n$ = 33,491/28,604 bouts from 74/78 fish for control/fin-amputated. **(D)** Righting gain is defined as the numeric inversion of the slope of the best fit line of rotation during deceleration vs. initial posture. $n$ = 121,979 bouts from 537 fish. **(E)** Steering gain of vestibular-impaired larvae vs. controls. *otog* mutation: $P_{t-test}$ = 2.33e-5; tangential lesions: $P_{t-test}$ = 7.41e-3; vestibulospinal lesions: $P_{t-test}$ = 2.32e-3. $N$ = 5/8/8 experimental repeats for *otog*/tangential lesions/vestibulospinal lesions. Same as follows. **(F)** Lifting gain of vestibular-impaired larvae vs. controls. *otog* mutation: $P_{t-test}$ = 7.09e-4; tangential lesions: $P_{t-test}$ = 3.57e-2; vestibulospinal lesions: $P_{t-test}$ = 5.90e-3. **(G)** Righting gain of vestibular-impaired larvae vs. controls. *otog* mutation: $P_{t-test}$ = 6.76e-4; tangential lesions: $P_{t-test}$ = 0.180; vestibulospinal lesions: $P_{t-test}$ = 1.00e-3. **(H)** Summary of effects of vestibular perturbations on bout kinematics. Vestibular-impaired fish swim with more eccentric posture and less fin-based lift. TAN, tangential. See also Table 1 for parameter definitions and Table 3 for statistics. All code and data can be found at DOI: 10.17605/OSF.IO/AER9F.
(TIF)

# Acknowledgments

The authors thank Christina May and other members of the Schoppik and Nagel laboratories for their valuable feedback and discussions.

## Author Contributions

**Conceptualization:** Yunlu Zhu, David Schoppik.

**Formal analysis:** Yunlu Zhu.

**Funding acquisition:** Yunlu Zhu, Kyla R. Hamling.

**Investigation:** Yunlu Zhu, Hannah Gelnaw, Franziska Auer, Kyla R. Hamling, David E. Ehrlich.

**Methodology:** Yunlu Zhu.

**Visualization:** Yunlu Zhu.

**Writing – original draft:** Yunlu Zhu.

**Writing – review & editing:** Yunlu Zhu, David Schoppik.

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
