## [Editor Report · Decision Letter 0]

13 May 2024

Dear David, 

It's great to hear from you!

Thank you for submitting your manuscript entitled "A brainstem circuit for gravity-guided vertical navigation" for consideration as a Short Reports by PLOS Biology.

Your manuscript has now been evaluated by the PLOS Biology editorial staff as well as by an academic editor with relevant expertise and I am writing to let you know that we would like to send your submission out for external peer review.

Once your full submission is complete, your paper will undergo a series of checks in preparation for peer review. After your manuscript has passed the checks it will be sent out for review. To provide the metadata for your submission, please Login to Editorial Manager (https://www.editorialmanager.com/pbiology) within two working days, i.e. by May 15 2024 11:59PM.

Kind regards,

Christian

Christian Schnell, PhD, 

Senior Editor

PLOS Biology

cschnell@plos.org

---

## [Decision Letter · Decision Letter 1]

13 Jun 2024

Dear David,

Thank you for your patience while your manuscript "A brainstem circuit for gravity-guided vertical navigation" was peer-reviewed at PLOS Biology. It has now been evaluated by the PLOS Biology editors, an Academic Editor with relevant expertise, and by several independent reviewers. 

In light of the reviews, which you will find at the end of this email, we would like to invite you to revise the work to thoroughly address the reviewers' reports.

As you will see below, the reviewers find that the topic of your study is interesting and overall think that it is well conducted. However, they also make a couple of comments where additional analyses are required or the language needs to be toned down. 

Given the extent of revision needed, we cannot make a decision about publication until we have seen the revised manuscript and your response to the reviewers' comments. Your revised manuscript is likely to be sent for further evaluation by all or a subset of the reviewers.

**IMPORTANT - SUBMITTING YOUR REVISION**

*Re-submission Checklist*

*Published Peer Review*

*PLOS Data Policy*

*Blot and Gel Data Policy*

Sincerely,

Christian

Christian Schnell, PhD

Senior Editor

PLOS Biology

cschnell@plos.org

REVIEWS:

Reviewer #1: The paper "A brainstem circuit for gravity-guided vertical navigation" by Zhu et al., investigates how larval zebrafish structure navigational movements in the depth direction. In larval zebrafish, turn directions and swim-speeds in the horizontal plane are structured rather than selected at random during both spontaneous exploration as well as navigation. While navigational strategies in the horizontal plane have been examined in many animal species, if and how movements are structured in the "z-direction" is far less explored. Given the importance of navigation in height or depth across species, this manuscript begins to address the critical question of how z-movements may be structured in order to allow for efficient exploration or navigation in depth. I strongly believe that this will be of great interest to the scientific community. The paper is very clear, well laid out and the methods used are highly appropriate for the study. I truly enjoyed reading it. While I was initially unsure whether the word "navigation" is appropriate in the title of the paper, considering that there does not seem to be a clear goal presented to the zebrafish, I do agree with the authors that a) navigation can be understood broadly as a means to accomplish movement in a particular direction and b) that the presented data strongly argues that the building blocks identified by the authors serve goal directed navigation as well. While I think that this manuscript should be published, I have one major reservation that has to be addressed.

*Major concern*

It is unclear what data the reported statistics are based on. Table 2 suggests that the statistics compare across fish (i.e. N=<number of fish>), however, the figure legends and methods are very ambiguous about this point. If statistics are indeed calculated across fish averages, this should be clarified in the figure legends (by stating the relevant N rather than both the number of fish and the number of swim bouts) and made explicit in the methods. If statistics are currently calculated across individual swim bouts, this should be corrected. Given that the manipulations carried out / used by the authors (ablations, mutations) are performed on a per-fish and not on a per-bout basis, the natural N would be the number of fish tested.

*Minor issue*

I would be curious to know at what point the correlation of consecutive bouts reaches chance level, i.e. becomes indistinguishable from shuffled controls. I understand that there might be a limitation of SAMPL in how long trajectories it can capture, however if it is possible to get at this point, I think it would make a worthwhile addition. Compared to angular correlations observed in the horizontal plane, these R^2 values are strikingly high, and knowing when they reach chance level could give a bound on streak length used during navigational movements.

While I'm generally fine with the use of the word "navigation" (see above), I do think that speaking of vertical navigation efficiency (first paragraph of the Discussion) is a bit misleading. Mainly, if there were a goal, maintaining consistent upward or downward trajectories once the goal has been reached would actually be highly inefficient as a navigational strategy.

Reviewer #2: The manuscript of Yunlu Zhu et al. titled "A brainstem circuit for gravity-guided vertical navigation" describes a circuit involved in navigation based on the perception of gravity. They use genetic and targeted manipulation to explore the role of a conserved brainstem circuit in the sensorimotor transformations involved in vertical navigation in zebrafish.

Overall, the manuscript is clear and well written, as are the figures. The data is well presented and easy to understand and follow. I am however concerned by the mildness of their manipulations' effect sizes, and the fact the data does not appear to support their conclusions. Despite mentions of crucial, indispensable, or essential, the data represents changes of 5-10 % in navigation efficiency. 

The experimental design is appropriate, and the statistical analysis correct. The tables recapitulate all the data very nicely and conveniently. The code provided seems adequate, although I have not tested it. The supplementary information are welcome, and the methods detailed. However, I do not think the results are provocative, or would have a general interest to a wider audience. The very strong causal language used throughout the manuscript is not supported by the data in my opinion, which is detrimental to the manuscript as a whole.

Major comments

There seems to be a high variability between experiments, and it is unclear if each comes from a different clutch? For example, the vestibulospinal neurons controls (S2) are worse than many of the manipulations in the other parts of the paper. 

The effect of the otog-/- and lack of utricle is surprisingly small (less than 5%) in 2B. A figure similar to 1K for the mutants would help assess their trajectory. Furthermore, it seems that the consistency of the WT in 1G is intermediary between the Hets and the Mut in 2C (which is confirmed by the very useful Table 1 and 2). The same with the efficacy, which is only 9.91e-2 for the WT in Table 1, compared to 9.05e-2 for the mutant. This value was deemed efficient on line 69, so it seems unclear how important the utricle truly is based on those results and how this data supports the "crucial" role of this otoliths in the observed behaviour.

What is the explanation for the increase in efficiency of the vestibulospinal-lesioned larvae? 

The observations of swim kinematics are interesting, consistent, and raise a lot of questions. It seems that the larvae with any manipulation have a defect in their posture and orientation in the gravity field. But at that stage of development, it is unlikely that the semicircular canals are functional. How would the swim kinematics be affected? I find the results of 3H difficult to reconcile with the opposite effects of TAN and Vestibulospinal lesions on vertical navigation (3C and S2C). This is discussed (161-167), but there is no speculation or attempt at a conclusion that would reconcile these observations. 

What could be the contribution of the swim bladder to those vertical navigation? Somato- and proprioception could explain how the fish still largely maintain vertical navigation without gravity perception. This is a short report, so I do not expect the authors to address this question, but having some hypotheses to follow on would be of interest.

Is there any effect of the manipulations on the hunting capabilities of the larvae? Would that affect their development over the 2 days of the experiment?

Minor comments

Line 161: First, it demonstrates…

Figure S1: It may help to guide readers by adding the schematics showing the inner-ear and vestibular pathway and where each lesion is on that scheme.

Reviewer #3: Zhu et. al. present clear data showing that the tangential nucleus and INC/nMLF constitute a circuit that receives vestibular input and plays a key role in vertical navigation. They establish an assay to observe vertical navigation in the water column, and measure important parameters of navigation, including consistency of bout direction and efficacy of depth change. The mutant and ablation data indicate that lesions to the tangential nucleus and INC/nMLF have effects similar to having no otolith, while vestibulospinal neuron ablations have a different phenotype. These findings lay important groundwork for further investigation into the circuits by which animals sense gravity and use it for navigation. However, I have a few comments that need to be addressed before publication: 

1. More explanation of why and how zebrafish larvae navigate in depth is needed in the Introduction. There was one sentence about this in the Discussion, but the authors should clarify earlier and more fully why it may be important for larvae to swim upwards and downwards.

2. In plots of Variability and Efficacy, e.g. Figure 2 B and E, the legend says data points are plotted, but it looks more like a cloud, and it is impossible to see the individual points to get an idea of the distribution of the data. Some kind of density or violin plots would help to illustrate the range of individual fish values. 

3. Efficacy is a useful metric to use to compare each mutant/lesion population with its control, but it is a bit abstract and hard to imagine what that corresponds to for an individual larva's navigation behavior. In Figure 2, it would be helpful to plot mutant and het depth change vs. first bout direction as in Figure 1L to see the difference in slopes. 

4. To further illustrate how gravity-blind larvae swim, I would like to see some representative trajectories of mutant and het fish, as in figure 1A, chosen from the mode of the distribution of efficacy in each population.

---

## [Decision Letter · Decision Letter 2]

26 Sep 2024

Dear David,

Thank you for your patience while we considered your revised manuscript "A brainstem circuit for gravity-guided vertical navigation" for publication as a Short Reports at PLOS Biology. This revised version of your manuscript has been evaluated by the PLOS Biology editors, the Academic Editor and two the original reviewers.

Based on the reviews and on our Academic Editor's assessment of your revision, we are likely to accept this manuscript for publication, provided you satisfactorily address the remaining reviewer comments and the following data and other policy-related requests.

* We would like to suggest a different title to improve readability/accuracy: "Evolutionarily-conserved brainstem architecture enables gravity-guided vertical navigation"

* Please add the links to the funding agencies in the Financial Disclosure statement in the manuscript details.

* Please include the full name of the IACUC/ethics committee that reviewed and approved the animal care and use protocol/permit/project license. Please also include an approval number.

* DATA POLICY:

Regardless of the method selected, please ensure that you provide the individual numerical values that underlie the summary data displayed in the following figure panels as they are essential for readers to assess your analysis and to reproduce it: 1O, 2CEF, 3CEF, 4CEF, S3CEF and S4EFG

* CODE POLICY

We expect to receive your revised manuscript within two weeks. 

*Published Peer Review History*

*Press*

Sincerely,

Christian

Christian Schnell, PhD

Senior Editor

cschnell@plos.org

PLOS Biology

Reviewer remarks:

Reviewer #1: My comments have been sufficiently addressed by the authors and I would be happy to see this paper published. I have a few small comments for the authors below.

Minor issues with the text:

Line 63: "The INC/nMLF receives ascending inputs; lesions there disrupred heading and navigation efficacy." <- Lesions within the INC/nMLF or specifically of the ascending inputs?

Line 106: "A course of trajectory with greater direction changes results in higher veering." <- ?

Line 141: "In addition swim directions,..." <- "to swim directions"

Line 174: "... while lack the utricular..." <- "while they lack"

Line 244: "...to indicate the how strongly fish..." <- "to indicate how strongly the fish"

Reviewer #2: It is worrying that the authors seem to think only interesting articles should be thoroughly reviewed. Now that the authors clarified the different experiments were done in different genetic backgrounds, the high variability makes more sense, it does show a wide range of natural WT behaviours which are all deemed effective by the authors, so I also appreciate the tempering of the claims regarding dramatic/crucial effects of their interventions. Overall, the authors have made all aspects of the manuscripts clearer and I am satisfied with their answers.

The added navigation figure for otog-/- is welcome and does show more random trajectories, and the impairment to depth navigation/changes appears a bit more clearly. I understand that there are biophysical constraints that limit the variability, my point was that if they were gravity blind, the veering would be more random, and so the variability higher. The article that the authors cite was not done in the dark, and the conclusion states that the small effects could be due to compensation strategies and training, so I am not sure it supports the small effects they observe. However, I am now reassured regarding the phenotype they observe.

I appreciate the context and new discussion surrounding the VS lesions. 

As the authors know, linear accelerometer alone cannot discriminate between tilt and translations, which the fish experience during each bout. As discussed in 10.1016/j.conb.2023.102776 zebrafish larvae may use the temporal dynamics of stimuli to discriminate between the two. And specificity was observed in the otolith organs in zebrafish (10.1038/s41467-022-35190-9), which could explain how the semicircular canals are not needed. But this may be out of the scope of the discussion in this manuscript.

Regarding other sensory modalities, and the claim that the fish only maintain trajectory is undermined by the efficacy of depth changes being only slightly affected in otog-/-. Although by their own arguments, it seems the efficacy is mostly an effect of motor circuits and not of gravity sensing. For the swim bladder, I was more thinking along the lines of what was observed in 10.1038/s41467-023-36682-y as a biomechanical factor in righting itself. But I appreciate that these would be outside of the scope of this manuscript.

---

## [Editor Report · Decision Letter 3]

17 Oct 2024

Dear David,

Thank you for the submission of your revised Short Reports "Evolutionarily-conserved brainstem architecture enables gravity-guided vertical navigation" for publication in PLOS Biology. On behalf of my colleagues and the Academic Editor, Tom Baden, I am pleased to say that we can in principle accept your manuscript for publication, provided you address any remaining formatting and reporting issues. These will be detailed in an email you should receive within 2-3 business days from our colleagues in the journal operations team; no action is required from you until then. Please note that we will not be able to formally accept your manuscript and schedule it for publication until you have completed any requested changes.

PRESS

Sincerely,

Christian 

Christian Schnell, PhD

Senior Editor

PLOS Biology

cschnell@plos.org